# An efficient graph generative model for navigating ultra-large combinatorial synthesis libraries

**Aryan Pedawi**
Atomwise Inc.
aryan@atomwise.com

**Paweł Gniewek**
Atomwise Inc.
pawel@atomwise.com

**Chaoyi Chang**
Atomwise Inc.
cchang373@atomwise.com

**Brandon M. Anderson**[*][†]
Atomwise Inc.
branderson@gmail.com

**Henry van den Bedem**[†]
Atomwise Inc.
UCSF, Dept. of Bioengineering & Therapeutic Sciences
vdbedem@atomwise.com

## Abstract

Virtual, make-on-demand chemical libraries have transformed early-stage drug discovery by unlocking vast, synthetically accessible regions of chemical space. Recent years have witnessed rapid growth in these libraries from millions to trillions of compounds, hiding undiscovered, potent hits for a variety of therapeutic targets. However, they are quickly approaching a size beyond that which permits explicit enumeration, presenting new challenges for virtual screening. To overcome these challenges, we propose the **C**ombinatorial **S**ynthesis **L**ibrary **V**ariational **A**uto-**E**ncoder (**CSLVAE**). The proposed generative model represents such libraries as a differentiable, hierarchically-organized database. Given a compound from the library, the molecular encoder constructs a query for retrieval, which is utilized by the molecular decoder to reconstruct the compound by first decoding its chemical reaction and subsequently decoding its reactants. Our design minimizes autoregression in the decoder, facilitating the generation of large, valid molecular graphs. Our method performs fast and parallel batch inference for ultra-large synthesis libraries, enabling a number of important applications in early-stage drug discovery. Compounds proposed by our method are guaranteed to be in the library, and thus synthetically and cost-effectively accessible. Importantly, CSLVAE can encode out-of-library compounds and search for in-library analogues. In experiments, we demonstrate the capabilities of the proposed method in the navigation of massive combinatorial synthesis libraries.

## 1 Introduction

Virtual high throughput screening (vHTS) [45] has gained significant traction in early-stage drug discovery, owing in no small part to make-on-demand chemical libraries utilizing a *combinatorial synthesis* construction. These combinatorial synthesis libraries (CSLs) enable access to ultra-large swaths of chemical space from a considerably smaller set of chemically accessible building blocks that can be combined according to known synthesis routines. In recent years, these libraries have grown from millions, to billions, and now to trillions of compounds [21, 32, 42, 55]. For example, the Enamine REadily AccessibLe (REAL) libraries [19] leverage off-the-shelf molecular building blocks and parallel synthesis, permitting lead times on the order of a few weeks and ushering in an era of ever-decreasing latency between *in silico* and *in vitro* high throughput screening.

---

[*]Work performed while at Atomwise Inc. Current affiliation is with Atomic AI.
[†]Equal senior contributions.

36th Conference on Neural Information Processing Systems (NeurIPS 2022).

As a result of the combinatorial explosion these constructions enable, early-stage drug discovery has now "crossed the Rubicon" into the non-enumerative regime. This presents new challenges for *in silico* hit discovery and optimization, which rely on screening explicitly enumerated compounds. These methods are ill suited to the non-enumerative setting, scaling linearly with the number of compounds, which motivates our interest in designing scalable approaches for navigating such libraries.

In this paper, we propose the **C**ombinatorial **S**ynthesis **L**ibrary **V**ariational **A**uto-**E**ncoder, or **CSLVAE** for short (pronounced like *c'est la vie*). CSLVAE is a graph-based generative model that exploits the structure of CSLs towards efficient navigation of the relevant chemical space. Our model learns a hierarchy of keys over the components of the library, and uses these keys to process queries for retrieval. The encoder processes a molecular graph and returns as output a query vector, which the decoder uses to retrieve the molecule through an efficient sequence of query-key comparisons that utilizes the hierarchical construction of CSLs, requiring minimal autoregression and admitting efficient parallelization.

The main contributions of this paper are as follows:

- We present a novel graph generative model which acts as a "neural database," providing random access to ultra-large, non-enumerable compound libraries. As such, our model is guaranteed to generate valid and cost-effectively accessible molecules.

- Our model overcomes challenges with long autoregressive chains in compound generation, improving scalability to large molecular graphs.

- Our model reduces the number of parameters ten-fold relative to comparable methods, and offers considerable improvements in computational complexity for searching through CSLs.

## 2   Related Work

**Virtual high throughput screening and enumeration**    Often, the first step in a vHTS campaign is preparing a compound library for subsequent use [1, 16]. While highly optimized compound sampling and scoring techniques have been developed [17], these approaches nevertheless rely on an exhaustively accessible library. An exception is the virtual synthon hierarchical enumeration screening (V-SYNTHES) approach [42], which leverages the modular nature of parallel synthesis libraries. However, by design, V-SYNTHES does not permit query-based random access. On the other hand, SpaceMACS [43] and SpaceLight [4] can provide query-based access to modular libraries by decomposing the query into fragments, and matching those by similarity search to synthons in the library. In parallel to these efforts, machine learning has received significant attention in vHTS: for predicting activity scores given docked conformations [14, 41, 54], predicting activity scores given a ligand and protein separately (undocked) [37, 53], and in improving or altogether replacing classical molecular docking with machine learning approaches [38, 48, 49].

**Deep learning approaches to molecular generation**    De novo drug design has assumed an increasingly prominent role in identifying novel chemical matter in drug discovery campaigns [9, 35, 47, 51]. The two dominant neural network-based paradigms for molecular generation are text-based and graph-based generative models. Early work in text-based generative models (also called chemical language models) applied recurrent neural networks to SMILES strings [15, 44]. Although these methods have shown a great deal of promise and spurred interest in molecular generation within the ML community, they are not guaranteed to produce valid SMILES strings. Approaches utilizing grammar constraints of SMILES notation have been proposed to improve validity [11, 28]; separately the recently proposed SELFIES notation [27, 36] guarantees validity and has seen increased adoption as such. In both cases, however, there remain known drawbacks to modeling with such text-based representations of chemical matter (e.g., surjectivity, similar molecular structures having possibly large edit distances).

For some applications, it is of interest to utilize generative models which can fit molecular databases, permitting the navigation of these databases via the fitted model. The ability of language models to fit molecular databases was investigated in a prior study [2] that applied deep language models to GDB-13 [6], a database of $10^9$ compounds formed by fully enumerating molecules up to 13 atoms of element types C, N, O, S, and Cl, subject to simple chemical stability and synthetic feasibility

rules. The authors trained on 0.1% of the total library and find that the model was capable of covering roughly 70% of compounds in the GDB-13 library. Furthermore, the language model they trained generates compounds not satisfying the GDB-13 construction in approximately 15% of cases.

Graph generative models have received significant attention in recent years as an alternative to their text-based counterparts. The earliest of these models focused on generating graphs of a constant size in a single shot [46] or generating graphs of arbitrary size autoregressively, one atom or bond at a time [31, 40, 57, 34]. These approaches also struggle to reliably produce chemically valid molecules and encounter difficulties with large molecular graphs.

In an effort to address both points, fragment-based graph generative models have been proposed and are growing in popularity [22, 23, 24, 26]. These models have the advantage of guaranteeing chemical validity by decomposing molecules into valid sub-components and explicitly disallowing actions which yield invalid combinations of fragments. Such explicit validity checks can be performed on every action, at the cost of additional computation. While other text- and graph-based generative models tend to struggle with large molecular graphs due to the long autoregression chains needed to produce them, fragment-based graph generative models require autoregression lengths on the order of the number of fragments that comprise a molecule. This can be significant when the fragments themselves contain many atoms.

However, some issues persist due to general difficulties with autoregressive graph generation. Unlike text-based models, in which there is less ambiguity in the autoregression order (e.g., tokens are typically decoded in left-to-right order), graphs have no such canonical node order, which presents challenges for graph-based autoencoders [31, 56]. Furthermore, although they require shorter autoregression chains than their counterparts, existing fragment-based graph generative models nevertheless require autoregression lengths that grow in the overall size of the molecule since autoregressive decoding cannot be effectively parallelized.

While fragment-based graph generative models and SELFIES-based language models each address the issue of chemical validity, there is the separate challenge of synthetic accessibility. Prior work has cast doubt on the synthetic feasibility of compounds proposed by many existing generative models [12], which can limit the practical utility of these models in drug discovery applications if not appropriately addressed. Subsequent work has attempted to improve on these shortcomings by (i) including explicit penalties for synthetic inaccessibility via a scoring function [18], (ii) limiting the model to fragments from known compounds [33, 39, 50], or (iii) inducing bias towards simple and known synthetic pathways [7, 8, 20].

## 3 Methodology

This section formalizes combinatorial synthesis libraries and the proposed model, CSLVAE. Figure 1 provides an illustration of the approach. Details on the architectures used for the various modules which comprise CSLVAE can be found in the Appendix.

### 3.1 Preliminaries

CSLs are composed of a set of multi-dimensional synthesis tables (Figure 1, Panel A). Each synthesis table describes a multi-component chemical *reaction*, which we denote by an index $t \in T$ (e.g., a natural number). Following [42], we use the term *R-group* to refer to a singular component in a reaction, and denote it by an index $r \in R$. A multi-component reaction takes several reactants (components) as input to produce a single molecule via chemical synthesis. We let $\psi : T \to \mathcal{P}(R)$ be the function which returns the set of R-groups $\psi(t) \subset R$ associated with a reaction $t$, where $\mathcal{P}(\cdot)$ denotes the power set function.

Each R-group is spanned by a possibly large number of molecular building blocks, called *synthons*, which can be utilized in the corresponding reaction. We denote a synthon by an index $s \in S$ and represent it with a molecular graph, $\mathcal{G}_s \in \mathscr{G}_S$. We note that a synthon can belong to multiple R-groups. For convenience, we use $\sigma : R \to \mathcal{P}(S)$ to represent the function which returns the set of synthons $\sigma(r) \subset S$ belonging to a particular R-group $r$.

A *product* $x \in X$ is a molecule that is synthesized according to a $k_t$-component reaction with R-groups $\psi(t) = (r_t^{(1)}, \ldots, r_t^{(k_t)})$ and a corresponding synthon tuple $u = (s_u^{(1)}, \ldots, s_u^{(k_t)})$, where

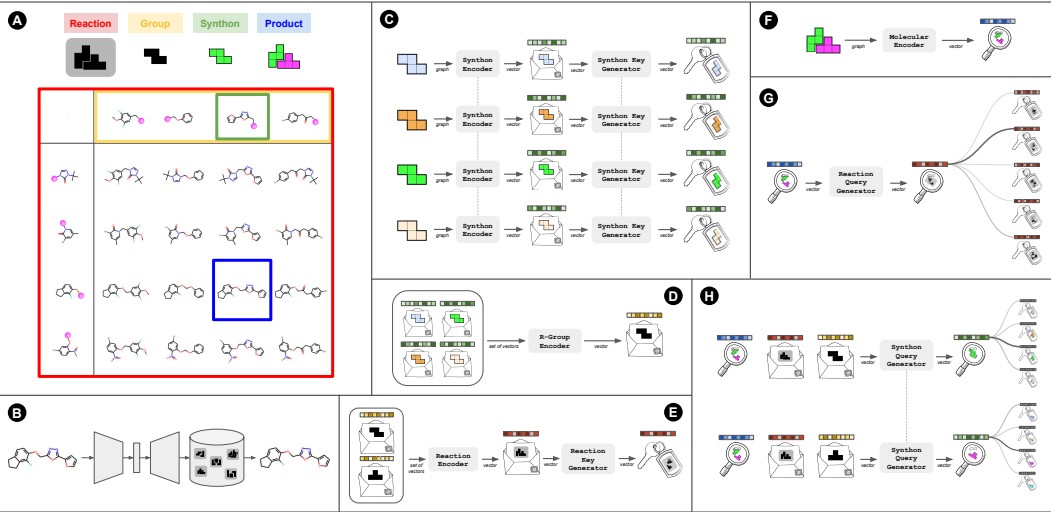

Figure 1: **Overview of CSLVAE. Panel A:** Illustration of a combinatorial synthesis table, which form the basis of combinatorial synthesis libraries (CSLs). **Panel B:** CSLVAE is an autoencoder whose encoder takes molecular graphs as input and returns query vectors as output, which the decoder then utilizes to retrieve the corresponding products from the library. **Panels C-E:** The CSLVAE library encoder represents CSLs via a hierarchy of learned representations. It proceeds by encoding individual synthons, represented as molecular graphs, with a graph neural network. It subsequently encodes R-groups with a set neural network over their constituent synthon representations. Finally, it encodes reactions with a set neural network over their constituent R-group representations. **Panel F:** The molecular encoder returns a query from an input molecular graph (in this case, a product from the library). **Panels G-H:** The molecular decoder processes the query to retrieve a product from the CSL, by first decoding the reaction type and subsequently decoding one synthon per R-group via query-key lookups.

$s_u^{(i)} \in \sigma(r_t^{(i)})$ for all $i = 1, \ldots, k_t$. We define $f$ to be the synthesis rule which generates a compound $x$ from a reaction and synthon tuple pair $(t, u)$, i.e., $x := f(t, u)$. In short, as a simple analogy, one can think of a synthesis rule $t$ as specifying an equation of $k_t$ terms, where each term is an R-group, and their associated synthons correspond to the allowed values for the corresponding term.

Hence, a combinatorial synthesis library $\mathcal{D} \equiv (T, R, S, f, \psi, \sigma)$ is fully characterized by its reactions $T$, R-groups $R$, and synthons $S$, together with the synthesis rule $f$, reaction to R-groups mapping $\psi$, and R-group to synthons mapping $\sigma$.

Using the language of probability, one can describe a distribution over products induced by $\mathcal{D}$ via the following factorization:

$$p(x|\mathcal{D}) = \sum_{t \in T} \sum_{u \in U_t} p(t|\mathcal{D}) \, p(u|t, \mathcal{D}) \, p_f(x|t, u), \tag{1}$$

where $U_t = \sigma(r_t^{(1)}) \times \ldots \times \sigma(r_t^{(k_t)})$ is the set of all eligible synthon tuples for reaction $t$. This factorization describes the generative process in which one first samples a reaction $t \sim p(t|\mathcal{D}) \propto |U_t|$, then samples a valid synthon tuple $u \sim p(u|t, \mathcal{D}) = |U_t|^{-1}$ comprised of synthons from the respective R-groups in $t$, and joins these together via synthesis to form a product $x$ (via the deterministic rule $f$).

As written, all valid $(t, u)$ pairs in $\mathcal{D}$ are equally probable under $p$. Note that if every product in $\mathcal{D}$ can be reached according to just a single synthesis route, then $p(x|\mathcal{D})$ is a uniform distribution over the part of chemical space $X$ accessible by $\mathcal{D}$.

### 3.2 Combinatorial Synthesis Library Variational Auto-Encoder (CSLVAE)

We consider the task of looking up a product $x$ from a library $\mathcal{D}$, which amounts to finding the reaction $t$ and synthon tuple $u$ satisfying $x = f(t, u)$. This can be cast as an inference problem

seeking $p(t, u|x, \mathcal{D})$. A latent variable model for $x$ gives rise to a variational formulation:

$$p(t, u|x, \mathcal{D}) = \int_{\mathbf{z}} p(t, u|\mathbf{z}, \mathcal{D})\, q(\mathbf{z}|x)\, d\mathbf{z}, \tag{2}$$

with $p(\mathbf{z})$ denoting the prior distribution of the latent variable. One can further simplify the joint conditional distribution of $t$ and $u$ by first selecting the reaction $t$ and then selecting the synthons $s_u^{(1)}, \ldots, s_u^{(k_t)}$ independently conditional on $t$:

$$p(t, u|\mathbf{z}, \mathcal{D}) = p(t|\mathbf{z}, \mathcal{D})\, p(u|\mathbf{z}, t, \mathcal{D}) \tag{3}$$

$$= p(t|\mathbf{z}, \mathcal{D}) \prod_{i=1}^{k_t} p(s_u^{(i)}|\mathbf{z}, t, r_t^{(i)}, \mathcal{D}). \tag{4}$$

This gives rise to a strategy in which one encodes a molecule $x$ into the latent space, $\mathbf{z} \sim q(\mathbf{z}|x)$, and proceeds by first decoding the reaction type $t \sim p(t|\mathbf{z}, \mathcal{D})$ and then, conditional on the sampled reaction, decoding one synthon per R-group $s_u^{(i)} \sim p(s_u^{(i)}|\mathbf{z}, t, r_t^{(i)}, \mathcal{D})$ for $i = 1, \ldots, k_t$ to form the synthon tuple $u \in U_t$.

The resulting latent variable model is the proposed CSLVAE. Panels B-H in Figure 1 give a step-by-step depiction of the method. In the forthcoming subsections, we will describe and formalize the three primary modules that comprise CSLVAE in detail: (i) the library encoder, (ii) the molecular encoder, and (iii) the molecular decoder. Module (ii) forms the basis for $q(\mathbf{z}|x)$, while modules (i) and (iii) form the basis for $p(t, u|\mathbf{z}, \mathcal{D})$.

### 3.2.1 Library encoder

A CSL $\mathcal{D} \equiv (T, R, S, f, \psi, \sigma)$ is organized hierarchically, with synthons $S$ at the bottom of the hierarchy, R-groups $R$ in the middle, and reactions $T$ at the top. We propose a strategy for learning an associated hierarchy of representations that describe the library at these three levels of resolution in an end-to-end fashion. These representations can then be used in retrieving results from queries into the library. The library encoder is illustrated in panels C-E of Figure 1.

We start from the bottom of the hierarchy with the synthons $S$. To learn a representation for each synthon in a manner that is fully inductive, we use a graph neural network to parameterize the $\texttt{SynthonEncoder} : \mathscr{G}_S \to \mathbb{R}^{d_S}$, which applies a sequence of message passing steps followed by a readout to arrive at a $d_S$-dimensional representation for each synthon.

Moving up the hierarchy, we use a deep set neural network to represent the R-groups $R$ by summarizing the representations for the synthons belonging to a particular R-group. Formally, the $\texttt{RGroupEncoder} : \mathbb{R}^{d_S} \times \cdots \times \mathbb{R}^{d_S} \to \mathbb{R}^{d_R}$ learns a $d_R$-dimensional representation for a given R-group from the set of $d_S$-dimensional representations of the constituent synthons. Following [58], the R-group encoder has the form $\texttt{RGroupEncoder}(\{\mathbf{h}_s^S : \forall s \in \sigma(r)\}) = \rho\left(\bigoplus_{s \in \sigma(r)} \phi(\mathbf{h}_s^S)\right)$, where $\phi$ and $\rho$ are parameterized by multi-layer perceptrons (MLPs) and $\bigoplus$ is a permutation-invariant aggregation operator. While other potentially more performant set-to-vector neural networks could be considered here, e.g., set transformers [30], an advantage of this simple construction is that it permits fast querying over library subsets at test time since $\phi(\mathbf{h}_s^S)$ can be cached for all $s \in S$, thereby reducing the required computations for making queries into partitions of the library. We utilize mean pooling as the aggregation operator to focus on characteristics of the distribution of synthons in an R-group (as opposed to sum pooling which generally expresses characteristics of the multiset), which improves performance when dealing with R-groups of varying cardinality.

Finally, to represent reactions, we use yet another deep set neural network as the $\texttt{ReactionEncoder} : \mathbb{R}^{d_R} \times \cdots \times \mathbb{R}^{d_R} \to \mathbb{R}^{d_T}$. This module takes as input a variable-sized set of R-group representations corresponding to the reactants in $t$ and produces a $d_T$-dimensional representation for the reaction. We utilize sum as the aggregation operator to learn multiset properties of the R-groups in a reaction.

Putting these together, the representations cascade as follows:

$$\mathbf{h}_s^S = \texttt{SynthonEncoder}(\mathcal{G}_s), \tag{5}$$

$$\mathbf{h}_r^R = \texttt{RGroupEncoder}(\{\mathbf{h}_s^S : \forall s \in \sigma(r)\}), \tag{6}$$

$$\mathbf{h}_t^T = \texttt{ReactionEncoder}(\{\mathbf{h}_r^R : \forall r \in \psi(t)\}). \tag{7}$$

In (4), we considered a factorization of the likelihood such that, given a molecular representation $\mathbf{z}$, the molecular decoder proceeds by first decoding the reaction type and then decoding one synthon for each R-group separately conditional on the reaction type. Hence, we require a key vector for each reaction as well as for each synthon to compare with the associated query vectors (to be described). As such, we introduce a `ReactionKeyGenerator` $: \mathbb{R}^{d_T} \to \mathbb{R}^{k_T}$ which returns a key vector given a reaction representation, and similarly introduce a `SynthonKeyGenerator` $: \mathbb{R}^{d_S} \to \mathbb{R}^{k_S}$ to produce a key for each synthon. Each of these key generators can be parameterized by MLPs:

$$\mathbf{k}_t^T = \texttt{ReactionKeyGenerator}(\mathbf{h}_t^T), \tag{8}$$

$$\mathbf{k}_s^S = \texttt{SynthonKeyGenerator}(\mathbf{h}_s^S). \tag{9}$$

### 3.2.2 Molecular encoder

The `MolecularEncoder` $: \mathscr{G}_X \to \mathbb{R}^{d_X}$ takes as input a molecular graph $\mathcal{G}_x \in \mathscr{G}_X$ and returns a $d_X$-dimensional feature representation,

$$\mathbf{z} = \texttt{MolecularEncoder}(\mathcal{G}_x). \tag{10}$$

In our implementation, the `MolecularEncoder` is a graph neural network with a variational linear layer stacked on top of the readout, which produces a sample $\mathbf{z} \sim q(\mathbf{z}|x)$ for a given input graph $\mathcal{G}_x$. As depicted in Figure 1, we interpret the representation $\mathbf{z}$ as a query induced by $x$ into the library $\mathcal{D}$.

Note that the `MolecularEncoder` can in principle take any valid molecular graph as input is therefore capable of producing queries for compounds that are *not* in the library $\mathcal{D}$. This can be useful for finding *analogues by catalog* – that is, compounds which can be purchased from a catalog and are chemical analogues of a query molecule. This use case will be investigated in a later section.

### 3.2.3 Molecular decoder

Given a query $\mathbf{z} = \texttt{MolecularEncoder}(\mathcal{G}_x)$ and a library $\mathcal{D}$, the decoder is tasked with retrieving the molecule from the library, i.e., identifying the reaction and synthon tuple which yield the molecule as a product. We proceed by generating the reaction first. A `ReactionQueryGenerator` $: \mathbb{R}^{d_X} \to \mathbb{R}^{k_T}$ generates a query from the molecular representation to compare against the reaction keys:

$$\mathbf{q}^T = \texttt{ReactionQueryGenerator}(\mathbf{z}), \tag{11}$$

$$\left\{ p(t_j|\mathbf{z}, \mathcal{D}) \right\}_{j=1}^{|T|} = \text{softmax} \left( \left\{ \frac{\mathbf{q}^T \cdot \mathbf{k}_{t_j}^T}{\sqrt{k_T}} \right\}_{j=1}^{|T|} \right). \tag{12}$$

This defines a probability distribution over reaction types in $T$. We can sample according to this distribution to arrive at a reaction $t \sim p(t|\mathbf{z}, \mathcal{D})$.

Given the sampled reaction $t$, we know the required R-groups (via $\psi$) and further know which synthons are eligible for each R-group (via $\sigma$). To decode the synthon tuple, we introduce a `SynthonQueryGenerator` $: \mathbb{R}^{d_X + d_T + d_R} \to \mathbb{R}^{k_S}$ which can be used to query synthons for each R-group $r_t^{(i)} \in \psi(t)$ as follows:

$$\mathbf{q}_{t,r}^S = \texttt{SynthonQueryGenerator}([\mathbf{z}\|\mathbf{h}_t^T\|\mathbf{h}_r^R]), \tag{13}$$

$$\left\{ p\big(s_j^{(i)}|\mathbf{z}, t, r_t^{(i)}, \mathcal{D}\big) \right\}_{j=1}^{|\sigma(r_t^{(i)})|} = \text{softmax} \left( \left\{ \frac{\mathbf{q}_{t,r_t^{(i)}}^S \cdot \mathbf{k}_{s_j^{(i)}}^S}{\sqrt{k_S}} \right\}_{j=1}^{|\sigma(r_t^{(i)})|} \right). \tag{14}$$

In our implementation, both the reaction and synthon query generators are parameterized by MLPs.

### 3.3 Training algorithm

For a large CSL $\mathcal{D}^*$, encoding the entire library on each iteration of the training loop could require an excessive amount of GPU memory. To overcome this, we utilize a minibatch strategy in which

a random subset $\mathcal{D} \subset \mathcal{D}^*$ is drawn from the full library according to a distribution $p(\mathcal{D}|\mathcal{D}^*)$. From $\mathcal{D}$, we form the synthon, R-group, and reaction representations and keys. In particular, we use a sub-sampler $p(\mathcal{D}|\mathcal{D}^*)$ which (i) samples a subset of the reactions contained in the full library uniformly at random, keeping only the R-groups contained in the sampled reactions, and (ii) for each reaction, samples a random number of products, retaining only the synthons that are contained in the sampled products. We also utilize teacher forcing, feeding in the ground truth reaction when generating the synthon queries for the respective R-groups. Algorithm 1 in the Appendix describes the training procedure.

**Ex-post density estimation** Given a trained generative model $p_\theta(x|\mathbf{z})$, we wish to sample products via $(x, \mathbf{z}) \sim p_\theta(x|\mathbf{z}) \, p(\mathbf{z})$ (discarding $\mathbf{z}$).[3] However, this in general will not correspond well to a uniform distribution over the products in $\mathcal{D}$ due to the bias introduced by the batch sampling strategy outlined above (which first uniformly samples reactions and then uniformly samples products given the reaction).

Although this can be corrected with importance weighting during the training phase, we opt for a more practical approach by using an ex-post density estimation strategy [13]. We sample a large number of products from the target distribution $x \sim p(x|\mathcal{D})$ and encode these products via the molecular encoder $\mathbf{z} \sim q_\phi(\mathbf{z}|x)$. We then fit a density estimator to the aggregated samples, written $q_\lambda(\mathbf{z})$. In our experiments, we utilize a multivariate normal distribution for simplicity, but one could imagine using more expressive density estimators here (e.g., a mixture of multivariate normals).

Now, we can sample products via $(x, \mathbf{z}) \sim p_\theta(x|\mathbf{z}) \, q_\lambda(\mathbf{z})$, which will more closely align with sampling from $p(x|\mathcal{D})$. This helps to correct for bias in the distribution over product space that is induced by the choice of batch sampling strategy.

## 3.4 Computational complexity, scalability, and efficiency

We now examine the computational complexity, scalability, and efficiency of the proposed method.

First, we note that the CSL $\mathcal{D}$ can be encoded with $O(|S| + |R| + |T|)$ complexity – the constant depends on the complexity of the synthon, R-group, and reaction encoders. Nonetheless, this is logarithmic in comparison to naively encoding each product in $\mathcal{D}$, which has $O(|\mathcal{D}|)$ complexity.

More noteworthy is the computational complexity of the molecular decoder. For clarity, let us consider a simplified $\mathcal{D}$ comprised of a single $k$-component reaction. Let $M_i$ denote the number of synthons for R-group $i = 1, \ldots, k$. Naively, a nearest neighbor lookup in $\mathcal{D}$ requires $O(\prod_{i=1}^k M_i)$ complexity. CSLVAE, on the other hand, performs the lookup over synthons in each R-group independently, which attains $O(\sum_{i=1}^k M_i)$ complexity: a logarithmic improvement. Hence, the proposed molecular decoder is highly suitable for ultra-large CSLs that are of interest in early-stage drug discovery.

Another advantage of CSLVAE's decoding strategy is that it relies only minimally on autoregression. In fact, we only ever need to do a single step of autoregression, irrespective of the size of the graph being generated (autoregression length of exactly two). As such, CSLVAE gracefully scales to large and variable-sized molecular graphs that follow a combinatorial synthesis construction.

Lastly, we point out that our method is guaranteed to generate chemically valid—and perhaps more importantly, synthetically accessible—molecular graphs without performing explicit validity checks. This compares favorably with prior work, in which the validity of each candidate action is verified at each step of the autoregression, with invalid actions excluded from the choice set. Although cheminformatics libraries like RDKit [29] have efficient C++ implementations for these checks, they nonetheless increase runtime rather significantly. Further, in the absence of explicit validity checks, these models have been shown to generate invalid molecular graphs at a markedly higher rate [22].

## 4 Experiments

This section covers some of our attempts to validate CSLVAE's performance and highlight its capabilities. We include additional supplementary experiments in the Appendix.

---

[3]The parameter $\theta$ represents the parameters for the modules written in `typewriter` font.

**Data** We demonstrate the capabilities of CSLVAE on the Enamine REAL library, which is comprised of 340K synthons and over one thousand reactions. The reactions in REAL range from two to four components and the number of synthons per R-group range from the single digits to tens of thousands. In total, the REAL library describes a chemical space of over 16 billion commercially available compounds.[4] Note that this 16 billion compound library is relatively small compared to over-trillion compound libraries that are commercially available today; we use this more modest library size as it makes comparisons to other approaches tractable.

**Training** During training, we sample subsets of the library as follows. Of the roughly 1300 reaction types in the REAL database, we first uniformly sample 20 reactions at random, and subsequently sample 100 products per reaction, including the associated synthons in the library subset. These library subsets therefore describe roughly 300K-1.5M compounds each, which is significantly smaller than the complete library of 16 billion compounds. See Algorithm 1 in the Appendix for details.

**Testing** For test-time inference, we decode with respect to the full library of 16 billion compounds. This constitutes a test-time distribution shift relative to training, but we observed that CSLVAE generalizes remarkably well to the full library without modifications. For completeness, we include an analysis of the test-time distribution shift in the Appendix (Supplementary Figure 2). In the forthcoming analyses, we share results when performing inference on the full library, as this is our primary objective.

### 4.1 Molecular reconstruction and generation

Table 1: Comparison of RationaleRL, JT-VAE, and CSLVAE (ours) on synthon-based generative modeling.

|  | JT-VAE | RationaleRL | CSLVAE (ours) |
|---|---|---|---|
| # Parameters | 4.7M | 3.4M | 380K |
| Validity | 100.0% | 100.0% | 100.0% |
| Uniqueness | 80.1% | 96.3% | 98.8% |
| Average likelihood | 18.7% | 62.3% | 72.4% |
| In-library proportion | 2.9% | 50.9% | 100.0% |

We compare CSLVAE against two state-of-the-art molecular graph generative models: JT-VAE [22] and RationaleRL [24]. All three models were trained from scratch on the Enamine REAL library. Details on the experimental setup and architecture can be found in the Appendix.

In JT-VAE, molecular graphs are represented by junction trees over chemical fragments. Decoding proceeds by first generating the junction tree in a depth-first manner, placing a fragment in each node, and then subsequently orienting the fragments to match attachment points. RationaleRL, on the other hand, takes as input a starting chemical fragment or *rationale*. The decoder's objective is to complete the molecule in an autoregressive fashion (one graph edit per step). In our setting, we take a product from the library and remove all but one synthon, treating the resulting graph as the starting rationale. Thus, RationaleRL is tasked with generating the missing synthon.

Table 1 summarizes the key findings. First, we note that our implementation of CSLVAE has roughly 10x fewer parameters than the two alternatives considered, owing to the inductive nature of the library encoder. All three methods achieve 100% chemical validity, but CSLVAE achieves this result without explicit validity checks. The average likelihood is computed by taking the average of per-compound reconstruction likelihoods across a large number of products sampled from the library. This is a measure of how well the model is capable of reconstructing the full molecular graph (i.e., on average, how likely are we to reproduce the query molecule via the decoder) and can also loosely be interpreted as a measure of coverage/reachability (i.e., what proportion of the library are we able to faithfully cover). Finally, we highlight the challenges existing graph generative models face when applied to ultra-large CSLs, namely that they struggle to reliably generate in-library compounds. For JT-VAE,

---

[4]We release a subset of the library alongside our code for reproducing these experiments and to foster further research in the machine learning community applied to combinatorial synthesis libraries: `https://github.com/AtomwiseInc/cslvae`. Data provided with permission from and attribution to Enamine Ltd.

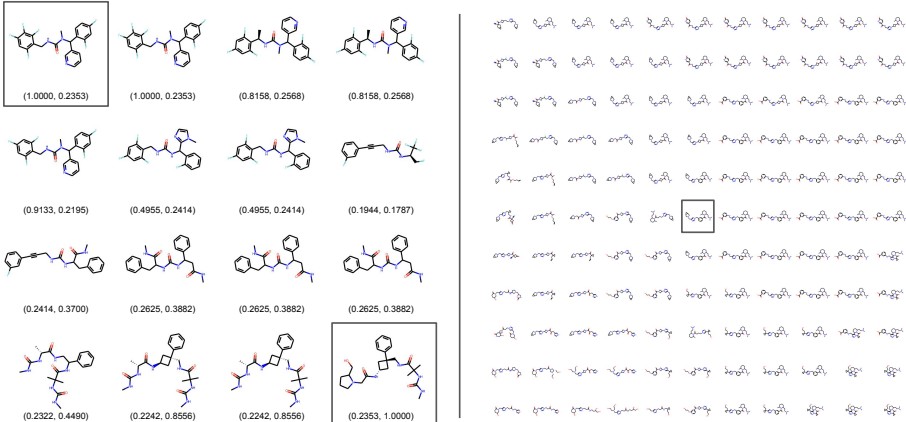

Figure 2: **Latent space visualizations. Panel A:** Moving left to right in raster order, we linearly interpolate from the starting compound to the target compound. The immediately adjacent molecules are reconstructions. Below each molecule are its Tanimoto similarities with the starting and target compound, respectively. **Panel B:** We sample two random directions in the latent space around a query compound and visualize the decoded molecules spaced evenly on the resulting 2D plane.

fewer than 1 in 34 compounds were found in REAL. RationaleRL, on the other hand, generates in-library completions in only about half of the cases (see Supplementary Figure 1 in the Appendix), but has the advantage that it is provided a starting rationale in the form of a compound from REAL stripped of all but one synthon. In contrast, CSLVAE is guaranteed to stay in the library by design.

## 4.2 Latent space visualizations

Next, we qualitatively inspect the latent space learned by CSLVAE. In particular, we are interested in verifying whether the proposed model has learned a latent space which varies relatively smoothly over the covered chemical space (i.e., that small perturbations to the query induce only minor edits in the resulting molecular graph). We perform two kinds of checks: latent space interpolations and local neighborhood visualizations.

Panel A of Figure 2 contains an example of interpolations in the latent space. In particular, we interpolate the molecular queries for the starting compound (top left) and target compound (bottom right), in raster order. The molecules immediately adjacent to the starting and target compounds are the associated reconstructions. Products are decoded with respect to the full REAL library of 16 billion compounds. Below each molecule are its Tanimoto similarity[5] with the starting and target molecule, respectively. We observe that the interpolations traverse through regions of chemical space that gradually decrease (cf. increase) in similarity with respect to the starting (cf. target) compound.

Panel B of Figure 2 visualizes the latent space around a randomly sampled product from the REAL library. Following prior work [28], we form a random 2D plane in the high-dimensional latent space by sampling two random directions around the molecular query (center compound) and decoding the resulting products using the argmax decision rule. We observe that the latent space is smooth in the sense that molecules morph gradually, with only minor edits when the movement in latent space is small (e.g., one synthon at a time, modifications to smaller functional groups), and that the molecular scaffold is generally conserved locally.

## 4.3 Analogue retrieval via autoencoding

Lastly, we utilize CSLVAE to find analogues of a query compound in a large CSL. In Figure 3, we present the model with two molecules: one which is in the library (left) and one which is not in the library (right). Given the molecular query, we generate conditionally random completions from the

---

[5]The Tanimoto similarity [3] is calculated by taking the intersection-over-union between a pair of bit vectors describing each molecule using a hash called a *molecular fingerprint* [5].

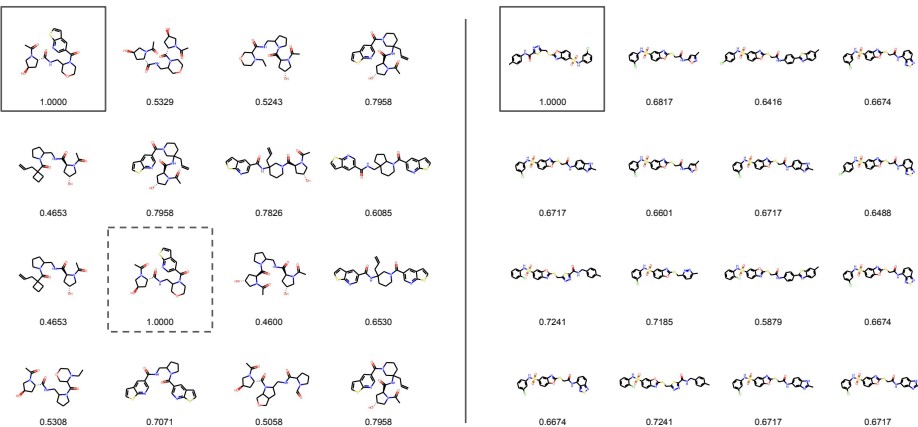

Figure 3: **Analogue retrieval via autoencoding.** Each query molecule (top left) is encoded and stochastically decoded fifteen times. Below each molecule is its Tanimoto similarity with the query molecule. **Left:** Autoencoding a molecule contained in the library. **Right:** Autoencoding a molecule not contained in the library.

decoder. In both cases, autoencoding retrieved highly similar compounds as measured by Tanimoto similarity. For the in-library example, we note that the model is able to successfully retrieve the query compound (third row, second column). Moreover, we find many instances in which autoencoding returns compounds that have the same reaction type as the query molecule, but vary in one or two synthons. For the out-of-library example, we note that CSLVAE finds skeletally relevant compounds with high Tanimoto similarity, indicating the promise of this approach for fast analogue search in large (non-enumerable) libraries.

## 5    Closing remarks

We proposed the combinatorial synthesis library variational auto-encoder, or CSLVAE, a new graph-based generative model for the navigation of combinatorial synthesis libraries. CSLVAE utilizes minimal autoregression, permitting efficient generation of large molecular graphs and improving scalability. Compounds generated by CSLVAE are chemically valid and cost-effectively accessible. CSLVAE is a neural database providing random access to non-enumerable libraries. In experiments, we demonstrate the capabilities of CSLVAE in modeling ultra-large and realistic make-on-demand libraries, paving a path towards more scalable strategies in the exploration of non-enumerable chemical libraries for early-stage drug discovery. Our method can be combined with established techniques for molecular optimization as a future direction.

Our approach has some limitations; here, we highlight three. First, the synthon lookup in the decoder scales linearly with the number of synthons in an R-group, which can present challenges as libraries continue to add many synthons per R-group. This could be mitigated with more scalable query-key designs [10, 25], but is not considered here. Furthermore, CSLVAE may face difficulties in the presence of prominent R-group symmetry (e.g., as in polymers). The decoder would require modifications to break parity, but may not admit the same convenient parallelization. Lastly, softmax has limitations in mapping from real-valued potentials to choice probabilities due to its rigid substitution patterns [52]; sparse or alternative-aware softmax variants may be preferable, but are not considered in this paper. In future work, we intend to address these shortcomings and look to applications in virtual high throughput screening.

## Acknowledgments and Disclosure of Funding

This work is sponsored by Atomwise Inc. The authors would like to give special acknowledgement to Christian Laggner, Ho Leung Ng, Srimukh Prasad, Adrian Stecula, and Brad Worley for discussions and helpful suggestions.

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
