# Appendix:
# An efficient graph generative model for navigating ultra-large combinatorial synthesis libraries

**Aryan Pedawi**
Atomwise Inc.
aryan@atomwise.com

**Paweł Gniewek**
Atomwise Inc.
pawel@atomwise.com

**Chaoyi Chang**
Atomwise Inc.
cchang373@atomwise.com

**Brandon M. Anderson**
Atomwise Inc.
branderson@gmail.com

**Henry van den Bedem**
Atomwise Inc.
UCSF, Dept. of Bioengineering & Therapeutic Sciences
vdbedem@atomwise.com

## A  CSLVAE details

This section provides additional details on CSLVAE omitted from the main paper due to the page limit.

### A.1  Training and ex-post density estimation algorithms

We describe the CSLVAE training procedure in Algorithm 1 and the CSLVAE ex-post density estimation procedure in Algorithm 2.

### A.2  Architecture

Below, we walk through the CSLVAE architecture utilized in our experiments. Supplementary Table 3 summarizes the overall CSLVAE architecture used in our experiments. In total, the version of CSLVAE we utilized is comprised of 384,512 learnable parameters.

#### A.2.1  Atom and bond embeddings

Both the synthon encoder and molecular encoder, which are parameterized as message passing graph neural networks, take shared atom and bond embeddings as the initial node and edge features, respectively. Atoms and bonds are represented by a set of binary features, which we described in Supplementary Tables 1 and 2, respectively. In our implementation, the node and edge dimensions are set to 64 each. As such, the atom embeddings consist of $50 \times 64 = 3{,}200$ parameters, and the bond embeddings consist of $12 \times 64 = 768$ parameters.

#### A.2.2  Synthon encoder

The `SynthonEncoder` is a message passing graph neural network, taking node, edge, and graph features as input, and producing updates to these features after every round of message passing in a residual fashion. We describe a single message passing layer below.

**Preliminaries**   A graph $\mathcal{G} = (\mathcal{V}, \mathcal{E})$ is represented by a set of node features $\{\mathbf{x}_i^0 : i \in \mathcal{V}\}$, edge features $\{\mathbf{e}_{ij}^0 : (i, j) \in \mathcal{E}\}$, and graph features $\mathbf{g}^0$. Superscripting by zero indicates that these are the initial or input features; below, superscripting by $\ell$ denotes the result after the $\ell$th round of message passing.

36th Conference on Neural Information Processing Systems (NeurIPS 2022).

**Algorithm 1** CSLVAE training procedure
---
**input** Full library $\mathcal{D}^*$, library sub-sampler $p(\mathcal{D}|\mathcal{D}^*)$, batch size $N$, model parameters $\theta$, KL divergence weight $\beta \geq 0$, choice of optimizer
**while** *stopping criteria not reached* **do**
    *# Prepare a minibatch*
    Sample a library $\mathcal{D} \sim p(\mathcal{D}|\mathcal{D}^*)$
    Sample a minibatch of reaction-synthon chain pairs, $(t_n, u_n) \sim p(t, u|\mathcal{D})$ for $n = 1 \ldots N$
    Form the corresponding products via synthesis, $x_n = f(t_n, u_n)$ for $n = 1 \ldots N$
    *# Encode the library*
    Get the synthon representations, $\mathbf{h}_s^S$ for each $s \in S$, as per equation (5)
    Get the R-group representations, $\mathbf{h}_r^R$ for each $r \in R$, as per equation (6)
    Get the reaction representations, $\mathbf{h}_t^T$ for each $t \in T$, as per equation (7)
    Get the reaction keys, $\mathbf{k}_t^T$ for each $t \in T$, as per equation (8)
    Get the synthon keys, $\mathbf{k}_s^S$ for each $s \in S$, as per equation (9)
    *# Encode the products*
    Sample the product queries, $\mathbf{z}_n$ for each $n = 1 \ldots N$, as per equation (10)
    Calculate the KL divergence contributions, $\mathrm{KLD}_n = \mathrm{D}_{\mathrm{KL}}(q(\mathbf{z}|x_n)\|p(\mathbf{z}))$ for each $n = 1 \ldots N$
    *# Decode the products wrt the library*
    Get the reaction queries, $[\mathbf{q}^T]_n$ for $n = 1 \ldots N$, as per equation (11)
    Get the reaction probabilities, $p(t|\mathbf{z}_n, \mathcal{D})$ for each $t \in T$ and $n = 1 \ldots N$, as per equation (12)
    Get the synthon queries, $[\mathbf{q}_{t_n,r}^S]_n$ for each $n = 1 \ldots N$ and $r \in \sigma(t_n)$, as per equation (13)
    Get the synthon probabilities, $p(s_j^{(i)}|\mathbf{z}_n, t_n, r_{t_n}^{(i)}, \mathcal{D})$ for each $n = 1 \ldots N$, $s_j^{(i)} \in \psi(r_{t_n}^{(i)})$, and
        $r_{t_n}^{(i)} \in \sigma(t_n)$, as per equation (14)
    Calculate the LL contributions, $\mathrm{LL}_n = \log p(t_n|\mathbf{z}_n, \mathcal{D}) + \sum_{i=1}^{|u_n|} \log p(u_n^{(i)}|\mathbf{z}_n, t_n, r_{t_n}^{(i)}, \mathcal{D})$ for
        each $n = 1 \ldots N$
    *# Compute the loss and update parameters*
    Calculate the loss, which is the negative of the weighted ELBO, $\ell = \frac{1}{N} \sum_{n=1}^{N} \beta \mathrm{KLD}_n - \mathrm{LL}_n$
    Compute gradients $\nabla_\theta \ell$ and update $\theta$ using choice of optimizer, i.e., $\theta \leftarrow \mathrm{optimizer}(\theta, \nabla_\theta \ell)$
**end while**
**return** Fitted model parameters $\theta^* \leftarrow \theta$
---

**Algorithm 2** CSLVAE ex-post density estimation procedure
---
**input** Master library $\mathcal{D}$, number of examples $N$, fitted model parameters $\theta^*$, density estimator $q_\lambda$
**for** $n = 1, \ldots, N$ **do**
    Sample $x_n \sim p(x|\mathcal{D})$, as per equation (1)
    Sample a product query $\mathbf{z}_n \sim q_{\theta^*}(\mathbf{z}|x_n)$ as per equation (10)
**end for**
Fit density $q_\lambda(\mathbf{z})$ to the aggregated product queries $\{\mathbf{z}_n\}_{n=1}^N$
**return** Fitted density estimator $q_{\lambda^*} \leftarrow q_\lambda$
---

**Edge model**    The edge model updates the edge features as follows. For an edge $(i, j) \in \mathcal{E}$, we concatenate the current iterate $\ell$ of the edge features, graph features, and the two corresponding node features. The concatenated features are then layer normalized and processed by a two-layer MLP, producing $\Delta \mathbf{e}_{ij}^{\ell+1} = \mathrm{Linear}(\mathrm{ReLU}(\mathrm{Linear}(\mathrm{ReLU}(\mathrm{LayerNorm}([\mathbf{e}_{ij}^\ell \| \mathbf{g}^\ell \| \mathbf{x}_i^\ell \| \mathbf{x}_i^\ell])))))$. The output dimension of the final linear layer is set to be equal to the dimensionality of the edge features, which allows the edge features to be updated in a residual fashion, i.e., $\mathbf{e}_{ij}^{\ell+1} = \mathbf{e}_{ij}^\ell + \Delta \mathbf{e}_{ij}^{\ell+1}$. Note that for an edge $(i, j) \in \mathcal{E}$, we separately maintain edge features for $i \rightarrow j$ and $i \leftarrow j$ updates; these features are identical for $\ell = 0$, but are not the same in general for subsequent layers.

**Node model**    Given a node $i \in \mathcal{V}$, we first form a message from all its neighboring nodes by summing over the associated incoming edge features, $\mathbf{m}_i^{\ell+1} = \sum_{j \in \mathcal{N}(i)} \mathbf{e}_{ij}^{\ell+1}$. Similar to the edge model, the node feature, message, and graph feature are concatenated, layer normalized, and passed through a two-layer MLP in which the final linear layer has output dimension equal to the dimensionality of the node features. This produces a residual update for the node feature, given

Supplementary Table 1: Atom feature types.

| Feature | Choices | Number of choices |
|---|---|---|
| Element type | *, B, Br, C, Cl, F, Fe, I, N, O, P, S, Se, Si, Sn | 15 |
| Node degree | 1, 2, 3, 4, 5, 6, $\geq 7$ | 7 |
| Hybridization | S, SP, SP2, SP3, SP3D, SP3D2, Unspecified | 7 |
| Chirality | CW, CCW, Other, Unspecified | 4 |
| Bonded hydrogens | 0, 1, 2, 3, 4, 5, 6, $\geq 7$ | 8 |
| Formal charge | $\leq$-3, -2, -1, 0, 1, 2, $\geq 3$ | 7 |
| Aromatic | False, True | 2 |
| | | 50 |

Supplementary Table 2: Bond feature types.

| Feature | Choices | Number of choices |
|---|---|---|
| Bond type | Single, Double, Triple, Aromatic | 4 |
| Conjugated | False, True | 2 |
| In a ring | False, True | 2 |
| Stereochemistry | One, Z, E, Any | 4 |
| | | 12 |

by $\Delta \mathbf{x}_i^{\ell+1} = \texttt{Linear}(\texttt{ReLU}(\texttt{Linear}(\texttt{ReLU}(\texttt{LayerNorm}([\mathbf{x}_i^{\ell} \| \mathbf{m}_i^{\ell+1} \| \mathbf{g}^{\ell}])))))$. The node features are thus updated according to $\mathbf{x}_i^{\ell+1} = \mathbf{x}_i^{\ell} + \Delta \mathbf{x}_i^{\ell+1}$.

**Graph model**  The graph features are updated by accumulating messages from each node in $\mathcal{V}$ as follows. Node messages to the graph $\{\mathbf{n}_i^{\ell+1} : i \in \mathcal{V}\}$ are formed using the now familiar layer norm + MLP design, with the final linear layer having output dimension equal to the dimensionality of the graph feature. These messages, $\mathbf{n}_i^{\ell+1} = \texttt{Linear}(\texttt{ReLU}(\texttt{Linear}(\texttt{ReLU}(\texttt{LayerNorm}([\mathbf{x}_i^{\ell} \| \mathbf{m}_i^{\ell+1} \| \mathbf{g}^{\ell}])))))$, are then aggregated via sum pooling to form a residual to the graph features, $\Delta \mathbf{g}^{\ell+1} = \sum_{i \in \mathcal{V}} \mathbf{n}_i^{\ell+1}$, which updates the graph features accordingly, $\mathbf{g}^{\ell+1} = \mathbf{g}^{\ell} + \Delta \mathbf{g}^{\ell+1}$. Implementation-wise, the node and graph model use a shared MLP with the output dimension equal to the sum of the node and graph feature dimensions, and the outputs are then split into two parts: one which routes to the node update, and the other to the graph update.

**Putting it all together**  The message passing neural network applies a sequence of message passing layers as outlined above. In our implementation, we set the node, edge, and graph feature dimensions to 64 and utilize four message passing layers. For the input graph features, we simply use a vector of zeros. The final graph features serve as the synthon representations, which are used in producing synthon keys and the cascaded representations for the R-groups and reactions. In total, the `SynthonEncoder` we utilize in experiments is described by 152,832 parameters.

### A.2.3  R-group encoder

The `RGroupEncoder` follows the design outlined in DeepSets, wherein the synthon representations are each processed separately by an MLP, pooled together (here, we use mean pooling to allow the network to focus on characteristics of the *distribution* of synthons belonging to an R-group), and the result processed by yet another MLP. In our experiments, both MLPs are set as two-layer networks with ReLU activation in between the two linear operations. All dimensions are 64, and as such the `RGroupEncoder` utilizes 16,640 parameters in total.

### A.2.4  Reaction encoder

The `ReactionEncoder` follows the same design as the `RGroupEncoder`, with the exception that sum pooling is utilized instead of mean pooling to allow the network to focus on the multi-set of R-groups in a reaction. All dimensions are 64, and as such the `ReactionEncoder` also utilizes 16,640 parameters in total.

### A.2.5 Synthon key generator

The `SynthonKeyGenerator` is an MLP which produces a synthon key from a synthon representation. In our implementation, we simply use a linear layer. The input and output dimensions are both set to 64, and as such the `SynthonKeyGenerator` utilizes 4,160 parameters in total.

### A.2.6 Reaction key generator

The `ReactionKeyGenerator` is an MLP which produces a reaction key from a reaction representation. In our implementation, we simply use a linear layer. The input and output dimensions are both set to 64, and as such the `ReactionKeyGenerator` utilizes 4,160 parameters in total.

### A.2.7 Molecular encoder

The `MolecularEncoder` produces molecular queries and utilizes the same message passing design as the `SynthonEncoder`, so refer to the early subsection describing the architecture. Again, the node, edge, and graph features are all set to 64, and we use four layers of message passing. However, differently from the `SynthonEncoder`, the `MolecularEncoder` has an additional variational linear layer that produces a conditional mean and conditional log variance vector from the graph features produced by the final message passing round. The variational linear layer takes a 64 dimensional graph feature as input and produces a 128 dimensional output, which is split into the mean and log variance portions. As such, the `MolecularEncoder` utilizes a total of 152,832 + 8,320 = 161,152 parameters.

## A.3 Molecular query processing network

The molecular queries are regularized to be close to the prior $p(\mathbf{z})$ as a consequence of the VAE objective. To allow the decoder to make better use of these features, the molecular queries are processed by an MLP prior to being passed to the reaction and synthon query generators. Opting for simplicity, we use a simple two-layer MLP with intermediate ReLU activation; all dimensions are 64, which results in 8,320 parameters.

### A.3.1 Reaction query generator

The `ReactionQueryGenerator` is an MLP which produces a reaction query from the molecular query. In our experiments, we utilize a two-layer network with intermediate ReLU activation. All dimensions are set to 64, so the `ReactionQueryGenerator` utilizes 8,320 parameters in total.

### A.3.2 Synthon query generator

Finally, the `SynthonQueryGenerator` is an MLP with produces a synthon query from the molecular query, reaction representation, and R-group representation. In the general case, we suggest concatenating these three feature types; because we have utilized a common dimensionality of 64 throughout and to keep implementation simple, we use sum instead of concatenation in our experiments (which can be shown to be equivalent to concatenating with an additional constraint on the weight matrix for the subsequent linear layer). Like the `ReactionQueryGenerator`, we also use a two-layer MLP with intermediate ReLU activation, resulting in a total of 8,320 parameters for the `SynthonQueryGenerator`.

## B Comparison of CSLVAE to RationaleRL and JT-VAE

In this section, we provide additional details on the experiment comparing CSLVAE with two state-of-the-art graph generative models, RationaleRL and JT-VAE, which is summarized by Table 1 in the body of the paper.

## B.1 Dataset preparation

The Enamine REAL library is a combinatorial synthesis library of roughly 1300 reaction types, ranging from 2- to 4-component reactions, along with roughly 340K synthons. In total, the REAL

Supplementary Table 3: Summary of CSLVAE architecture used in experiments.

| Module | Module type | Number of parameters |
|---|---|---|
| Atom embedding | Embedding | 3,200 |
| Bond embedding | Embedding | 768 |
| Synthon encoder | GNN | 152,832 |
| R-group encoder | DeepSets | 16,640 |
| Reaction encoder | DeepSets | 16,640 |
| Synthon key generator | Linear | 4,160 |
| Reaction key generator | Linear | 4,160 |
| Molecular encoder | GNN | 161,152 |
| Molecular query processing network | MLP | 8,320 |
| Reaction query generator | MLP | 8,320 |
| Synthon query generator | MLP | 8,320 |
| | | 384,512 |

library describes a chemical space in excess of 16B make-on-demand compounds. For the two baseline models we compare against, we faced memory issues when using the author-provided code on the full REAL library (both in the vocabulary generation steps as well as with writing products to disk). As such, the two baselines were trained and evaluated on a subset of the full REAL library. We did not face such memory limitations with CSLVAE, which we trained on the full library and evaluated as such. Hence, RationaleRL and JT-VAE can be compared on a per-item basis (each having been trained on the same subset of REAL), whereas our results for CSLVAE reflect training on the larger and more diverse full REAL library. We note that RationaleRL and JT-VAE were not developed with the goal of searching through combinatorial synthesis libraries in mind, and our use of them serves as an attempt to compare our method with the application of existing state-of-the-art graph generative model on combinatorial synthesis libraries out-of-the-box and without modification.

To construct the data on which RationaleRL and JT-VAE were trained, we ranked the ∼1300 reaction types by the number of products contained in each reaction and selected 50 of the middle-sized reactions. In total, these reactions describe a chemical space of ∼125M compounds. We sampled products from each reaction such that all synthons are represented, amounting to a training set of ∼500K compounds.

## B.2 RationaleRL details

We utilize the *pre-training* phase of RationaleRL, which trains a graph-based variational autoencoder that seeks to reconstruct a molecular graph from a *starting rationale* (and does not require RL, as the name might suggest; that is part of the *fine-tuning* phase, which we do not utilize here). Given the starting rationale and full molecule, RationaleRL's decoder completes the molecule autoregressively in an atom-by-atom, bond-by-bond fashion. To form the starting rationale, we take a product from REAL and remove all but one synthon. Hence, RationaleRL is tasked with completing the missing synthon given the full molecule (as input to the encoder) and the starting rationale (as input to the decoder, along with the latent code). In Supplementary Figure 1, we show that as the number of distinct reactions in training is increased, the probability that completions generated by RationaleRL yield a product from the library drops precipitously; after 50 reactions are included, fewer than half of all completions yield a product in the REAL library. Our experiments are based on the author-provided code, which can be found at `https://github.com/wengong-jin/multiobj-rationale`.

## B.3 JT-VAE details

JT-VAE is a graph-based variational autoencoder which generates molecules according to a tree-structured scaffold of fragments. Unlike RationaleRL, JT-VAE samples chemical *fragments* in an autoregressive fashion, rather than atoms or bonds. Prior to training, we apply the vocabulary generation step described in the JT-VAE paper to the sampled products from the subset of REAL utilized in the baseline experiments, which yields a total of 325 fragments. Our experiments are based on the author-provided code, which can be found at `https://github.com/wengong-jin/icml18-jtnn`.

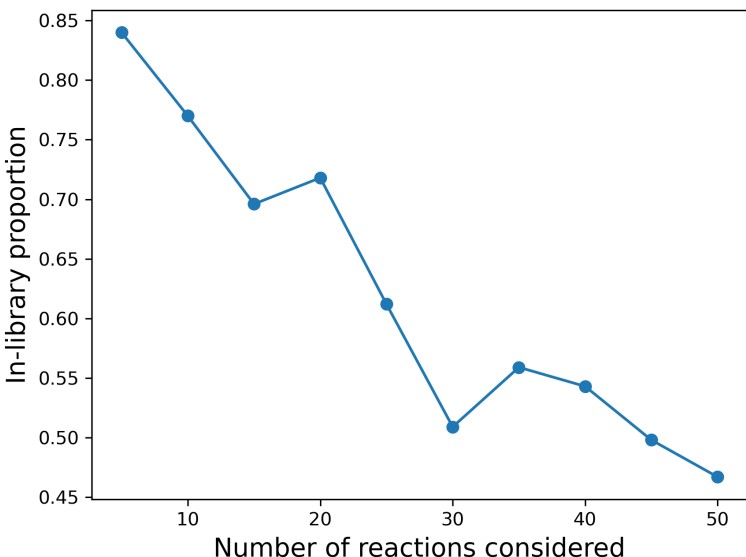

Supplementary Figure 1: As the number of distinct reactions present in training increases, the in-library proportion for generated samples from RationaleRL quickly decreases, dropping to below 50% after just 50 reactions are included.

### B.4 CSLVAE details

During training we utilize an annealing schedule on $\beta$ [3] (see Algorithm 1), starting with $\beta = 0$ and incrementing by 1e-5 every 2000 iterations, with a max value of $\beta = 1$. We train for a total of 200K iterations, in which time CSLVAE has seen a total of $2000 \times 200\text{K} = 400\text{M}$ compounds (although not 400M unique compounds, due to the batch sampling strategy). As such, by the time training is halted, CSLVAE has seen no more than 2.5% of the full REAL library.

## C   Test-time distribution shift in CSLVAE

As described in Algorithm 1, we sample small subsets of the full library in each training iteration for tractability. In particular, the library sub-sampler we utilize first samples uniformly over the reactions and subsequently samples a constant number of products in each reaction at random; the synthons associated with these sampled products comprise the synthons in the library subset. In training the CSLVAE model used in our experiments, we sample 20 reactions and 100 products per reaction, which yields a minibatch of 2000 products. As described in the paper, these associated library subsets describe a chemical space of roughly 300K-1.5M compounds each. At test time, however, we wish to decode according to the full library of 16B compounds, which constitutes a fairly drastic test-time distribution shift.

Supplementary Figure 2 illustrates the magnitude of the test-time distribution shift on reconstruction quality (measured by average likelihood). We start by sampling five library subsets as described above, drawing 2000 compounds for each subset; each of these five draws is represented by a different color in the figure. We then grow these library subsets by first including all synthons across the 20 sampled reactions, and calculate the average reconstruction likelihood for the same 2000 products according to the expanded library, and further increase the library by adding all synthons for 100 randomly sampled additional reactions, and then all synthons for yet another 100 randomly sampled additional reactions, and finally include all synthons and all reactions. At each step, the library subsets describe an increasingly larger chemical space. We observe that the average likelihood decreases in a roughly linear-log fashion in the library size as a result of this distribution shift.

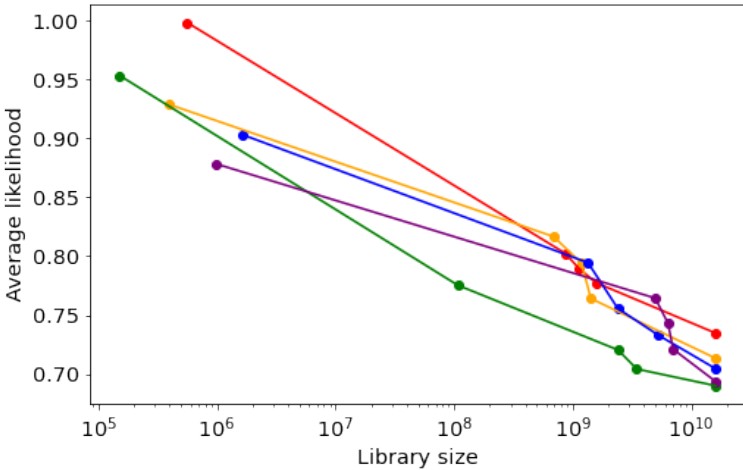

Supplementary Figure 2: Test-time distribution shift as a function of library size.

## D    Ex-post density estimation

In Section 3.3 of the paper, we describe the use of ex-post density estimation on latent codes corresponding to products sampled uniformly from the library as a way to force random samples from CSLVAE to track more closely to sampling uniformly at random from the library. Algorithm 2 outlines this procedure.

We carry out a simple experiment to demonstrate that this procedure achieves the intended result. The experiment proceeds as follows. We draw 10,000 molecules from the library uniformly at random which we treat as a training set for the ex-post density estimator. We consider three density estimators of increasing expressivity: a multivariate normal (MVNormal), a mixture of five normals (MoG-5), and a mixture of ten normals (MoG-10). We also compare to the standard approach of sampling from the isotropic multivariate normal prior. As such, we have four alternative schemes for sampling latent codes which are subsequently decoded into compounds from the library. We then sample another 10,000 molecules from the library, again uniformly at random, which is treated as a reference set for comparison.

Supplementary Table 4: Divergences between query sampling strategies and a reference set of compounds sampled uniformly at random from the library.

|            | Train  | Prior  | MVNormal | MoG-5  | MoG-10 |
|------------|--------|--------|----------|--------|--------|
| Uniqueness | 100%   | 96.9%  | 99.8%    | 99.7%  | 99.8%  |
| SA (JSD)   | 0.0179 | 0.3340 | 0.0651   | 0.0557 | 0.0452 |
| QED (JSD)  | 0.0134 | 0.2200 | 0.0572   | 0.0266 | 0.0225 |
| MW (JSD)   | 0.0169 | 0.4370 | 0.0970   | 0.0650 | 0.0686 |
| logP (JSD) | 0.0144 | 0.0259 | 0.0643   | 0.0686 | 0.0703 |

Following [2], we compare the aforementioned sets of generated molecules to the reference set of molecules on the following computable molecular properties: synthetic accessibility (SA), quantitative estimation of drug-likeness (QED), molecular weight (MW), and logarithm of the octanol-water partition coefficient (logP). In particular, we calculate the Jensen-Shannon distance (square root of the Jensen-Shannon divergence) between the distribution of these properties on the reference set and each set in question. Supplementary Table 4 summarizes the results of this exercise. Three items worth noting are that (a) the reference compounds and the compounds sampled for training the ex-post density estimators have low divergence across the various properties, (b) sampling from the prior generates compounds with high divergence relative to uniform sampling over the library, and (c) using more expressive density estimators for the latent codes leads to increasingly lower divergence

across the various properties with the reference set, as they are able to better match the distribution of latent codes for the training set (which is exchangeable with the reference set).

# E   Comparison to existing analogue enumeration approaches

We attempt a comparison of CSLVAE's analoguing capabilities with that of Arthor, a state-of-the-art commercial similarity search tool for synthesis libraries developed by NextMove Software. Arthor performs analogue enumeration using a custom ECFP4 bit vector representation of molecules, returning compounds with high Tanimoto similarity according to this fingerprint. For a given query compound, we enumerate the top-100 analogues returned by Arthor from REAL. Similarly, we encode each query compound with the CSLVAE encoder and generate a corresponding 100 stochastic decodings. For every analogue, we compute its RDKit ECFP4 Tanimoto similarity with the query compound and retain only the top-1 analogue. We then compare the distribution of Tanimoto similarities of the top-1 analogues returned in this way between Arthor and CSLVAE. As a control, we use a naive random baseline policy which samples 100 compounds at random from REAL, again selecting the top-1 analogue based on RDKit ECFP4 Tanimoto similarity. Because both CSLVAE and the random baseline constitute stochastic policies, we repeat this procedure 30 times for each query compound and take the average top-1 Tanimoto similarity.

Supplementary Table 5: List of FDA novel drug approvals in 2021 used in analogue comparison.

| Molecule | Canonical SMILES |
|---|---|
| asciminib | O=C(Nc1ccc(OC(F)(F)Cl)cc1)c1cnc(N2CCC(O)C2)c(-c2ccn[nH]2)c1 |
| atogepant | CC1C(c2c(F)ccc(F)c2F)CC(NC(=O)c2cnc3c(c2)CC2(C3)C(=O)Nc3ncccc32)C(=O)N1CC(F)(F)F |
| avacopan | Cc1ccc(NC(=O)C2CCCN(C(=O)c3c(C)cccc3F)C2c2ccc(NC3CCCC3)cc2)cc1C(F)(F)F |
| belumosudil | CC(C)NC(=O)COc1cccc(-c2nc(Nc3ccc4[nH]ncc4c3)c3ccccc3n2)c1 |
| belzutifan | CS(=O)(=O)c1ccc(Oc2cc(F)cc(C#N)c2)c2c1C(O)C(F)C2F |
| cabotegravir | CC1COC2Cn3cc(C(=O)NCc4ccc(F)cc4F)c(=O)c(O)c3C(=O)N12 |
| casimersen | CN(C)P(=O)(OC[C@@H]1CNC[C@H](n2ccc(=N)nc2O)O1)N1CCN(C(=O)OCCOCCOCCO)CC1 |
| drospirenone | CC12CCC3c4ccc(O)cc4CCC3C1C(O)C(O)C2O |
| finerenone | CCOc1ncc(C)c2c1C(c1ccc(C#N)cc1OC)C(C(N)=O)=C(C)N2 |
| fosdenopterin | Nc1nc(=O)c2c([nH]1)N[C@@H]1O[C@@H]3COP(=O)(O)O[C@@H]3C(O)(O)[C@@H]1N2 |
| maralixibat | CCCCC1(CCCC)CS(=O)(=O)c2ccc(N(C)C)cc2C(c2ccc(OCc3ccc(C[N+]45CCN(CC4)CC5)cc3)cc2)C1O |
| maribavir | CC(C)Nc1nc2cc(Cl)c(Cl)cc2n1[C@H]1O[C@@H](CO)[C@H](O)[C@@H]1O |
| mobocertinib | C=CC(=O)Nc1cc(Nc2ncc(C(=O)OC(C)C)c(-c3cn(C)c4ccccc34)n2)c(OC)cc1N(C)CCN(C)C |
| piflufolastat | O=C(O)CCC(NC(=O)NC(CCCCNC(=O)c1ccc(F)nc1)C(=O)O)C(=O)O |
| ponesimod | CCCN=C1SC(=Cc2ccc(OCC(O)CO)c(Cl)c2)C(=O)N1c1ccccc1C |
| samidorphan | NC(=O)c1ccc2c(c1O)C13CCN(CC4CC4)C(C2)C1(O)CCC(=O)C3 |
| serdexmethylphenidate | COC(=O)C(c1ccccc1)C1CCCCN1C(=O)OC[n+]1cccc(C(=O)NC(CO)C(=O)[O-])c1 |
| sotorasib | C=CC(=O)N1CCN(c2nc(=O)n(-c3c(C)ccnc3C(C)C)c3nc(-c4c(O)cccc4F)c(F)cc23)C(C)C1 |
| tepotinib | CN1CCC(COc2cnc(-c3cccc(Cn4nc(-c5cccc(C#N)c5)ccc4=O)c3)nc2)CC1 |
| tivozanib | COc1cc2nccc(Oc3ccc(NC(=O)Nc4cc(C)on4)c(Cl)c3)c2cc1OC |
| trilacicilib | CN1CCN(c2ccc(Nc3ncc4cc5n(c4n3)C3(CCCC3)CNC5=O)nc2)CC1 |
| umbralisib | CC(C)Oc1ccc(-c2nn(C(C)c3oc4ccc(F)cc4c(=O)c3-c3cccc(F)c3)c3ncnc(N)c23)cc1F |
| vericiguat | COC(=O)Nc1c(N)nc(-c2nn(Cc3ccccc3F)c3ncc(F)cc23)nc1N |
| viloxazine | CCOc1ccccc1OCC1CNCCO1 |

For the query compounds, we use 24 of the 51 novel drugs approved in 2021 by the FDA, filtering out drugs which do not satisfy a routine set of small molecule criteria (such as monoclonal antibodies); the compounds used in this experiment are listed in Supplementary Table 5. Supplementary Figure 3 shows a boxplot of the top-1 Tanimoto similarities for Arthor, CSLVAE, and the random baseline. While CSLVAE finds more distant ECFP4 analogues compared to Arthor (which is a gold standard for fingerprint similarity search), it is nonetheless able to identify analogues[1] for unseen, novel drugs in a routine manner.

Of particular note, CSLVAE is considerably less resource intensive than Arthor, which is specially designed for fast and efficient fingerprint-based analogue enumeration and requires appropriate

---

[1] It is typical to use a Tanimoto similarity threshold in the range 0.3-0.35 to indicate whether a pair of molecules can be seen as analogues.

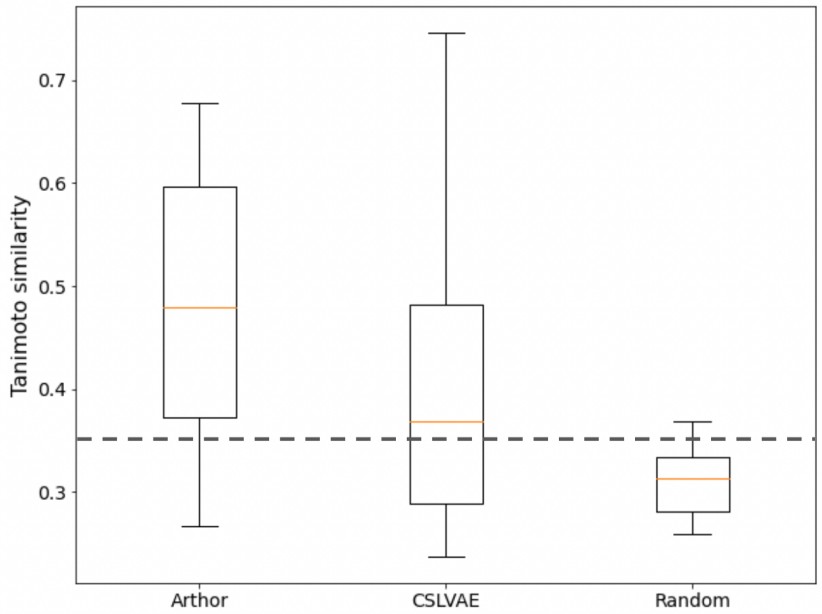

Supplementary Figure 3: Comparing CSLVAE to Arthor in top-1 analogue search.

infrastructure and setup. For these reasons, it is difficult to do a direct apples-to-apples comparison of the compute requirements. Nevertheless, we share the following pertinent information. The CSLVAE experiments were run on a machine with an NVIDIA Tesla K80 GPU and an Intel Xeon E5-2686 CPU. On average, sampling 100 analogues for a given query using CSLVAE completed in 11.41 seconds with this setup (encoding and decoding). Further, all of the parameters and buffers (including representations and keys for the synthon, R-group, and reactions) of the trained CSLVAE model required just 170MB memory. By comparison, the Arthor experiments are run in a distributed fashion on 100 pods, each with 8 CPUs, with top-100 analogue enumeration requiring a total of 32.46 seconds, for an approximate total CPU time of 25,968 seconds. Furthermore, Arthor needs to devote rather significant amounts of memory and storage to carry out analogue enumeration; our in-house setup uses 3GB per shard and makes 64GB RAM requests for each Arthor worker.

We note that CSLVAE's ability to represent large CSLs with significantly fewer resources and perform analogue retrieval with notably improved execution time (perhaps three orders of magnitude faster) owes to its decoding strategy, which utilizes parallel synthon look-ups, thereby requiring a number of keys that is on the order of number of synthons in the library rather than on the order of number of products in the library, and further permits a kind of similarity search in time that is logarithmic in the number of products in the library (rather than linear).

As a final remark, we view this exercise as a demonstration of CSLVAE's potential as a backbone in search applications, not as a definitive implementation of an analogue enumeration strategy with CSLVAE. There are a number of requirements of a good analogue enumeration tool that we purposefully do not focus on here (e.g., enumerating $k$ distinct analogues rather than top-1 analogue, where $k$ could be large). While this may be of interest in subsequent work, it is not our focus in this paper. We believe that the quality of analogues found with CSLVAE-based decoders can begin to approach those found by existing, specially-tailored tools for analogue enumeration with additional efforts aimed at such applications, and that such directions could prove fruitful in an era of combinatorial library explosion that challenges traditional enumerative strategies.

## F   Encoder transfer to molecular property prediction

The CSLVAE training objective can be viewed as a kind of contrastive pretext task, which seeks to align the representation of a given molecule with representations that correspond to retrieval instructions in a CSL. Hence, it is natural to wonder whether the encoder of a trained CSLVAE

model could demonstrate good transfer performance in prediction tasks that may be of interest. To investigate, we compare MLPs trained on CSLVAE with those trained on molecular fingerprints (ECFP4 and ECFP6) for molecular property prediction tasks. We use the octanol-water partition coefficient (logP) and the quantitative estimate of drug likeness (QED) [1] as targets for prediction.

Supplementary Table 6: Encoder transfer on logP and QED prediction. The cells report the average RMSE ± one standard deviation, calculated over five runs.

| | Dimensionality | REAL 100K logP | ZINC 250K logP | REAL 100K QED | ZINC 250K QED |
|---|---|---|---|---|---|
| CSLVAE | 64 | **0.539 ± 0.002** | 0.591 ± 0.001 | **0.072 ± 0.001** | 0.068 ± 0.001 |
| ECFP4 | 256 | 0.827 ± 0.001 | 0.679 ± 0.002 | 0.091 ± 0.002 | 0.079 ± 0.001 |
| ECFP6 | 1024 | 0.601 ± 0.002 | **0.490 ± 0.001** | **0.072 ± 0.001** | **0.064 ± 0.001** |

For this experiment, we construct a dataset by sampling 100K compounds uniformly at random from REAL, splitting the examples into training, validation, and testing folds using an 80-10-10 split. For each compound, we extract as feature descriptor its CSLVAE query, in addition to its ECFP4 and ECFP6 fingerprints. Using the training fold, we fit an MLP on each of these feature descriptors separately to predict the molecule's logP and QED score. We select the iteration which attains the lowest validation RMSE, recording its test RMSE. This is repeated five times, and we report the average test RMSE and standard deviation. To demonstrate the extent to which CSLVAE learns molecular features that are predictive of such molecular properties for out-of-domain compounds, we repeat this exercise on a dataset of 250K molecules from ZINC. The results of this exercise are summarized in Supplementary Table 6.

The results of this experiment confirm that the latent space learned by CSLVAE can indeed be utilized to success in predicting quantities like logP and QED, especially in the in-domain case where it out-performs the predictors fit on chemical fingerprints. However, in the out-of-domain case, the predictor fit on ECFP6 fingerprints performs notably better than the predictor fit on CSLVAE queries, suggesting that the features learned by CSLVAE may be missing some pertinent predictive information about input molecules which differ significantly from the CSL on which it was trained.