# OpenReview forum: "An efficient graph generative model for navigating ultra-large combinatorial synthesis libraries"
_NeurIPS.cc/2022/Conference — NeurIPS 2022 Accept_

### Official Review · Reviewer_UvPu · 2022-06-29

**Rating:** 6
**Confidence:** 3
**Soundness:** 2 fair
**Presentation:** 2 fair
**Contribution:** 3 good

**Summary:**

The present paper is concerned about learning a neural representation of a combinatorial synthesis library, whose size is now exploding and which is difficult to use for virtual screening due to its massive size.

The idea is to construct a generative model of a molecule that receives a continuous vector and outputs a molecule that is constructed by using the combinatorial synthesis library.
In particular, the proposed model named CSLVAE consists of a library encoder, a molecular encoder, and a molecular decoder.
The library encoder is used to encode the combinatorial synthesis library into keys of reactions and synthons, which are used to retrieve them during generation. The molecular encoder receives a molecular graph and outputs a continuous feature representation of it. The molecular decoder receives a continuous feature vector and constructs a molecule by first retrieving a reaction using the query constructed from the feature vector and the keys defined by the library encoder, and then, retrieving synthons in a similar way.

The authors empirically compare CSLVAE with baseline methods to show that CSLVAE can generate molecules in the library, whereas the baseline methods cannot. The authors also conduct qualitative analyses to discuss the benefit of CSLVAE.

**Questions:**

1. I would appreciate it if the authors could provide more detailed explanations in the preliminaries section.

**Limitations:**

The authors discussed several limitations of the proposed method in terms of computational efficiency, which are reasonable.

**Strengths And Weaknesses:**

# Strengths
- Originality: The problem setting to construct a neural database is interesting. While most of the researchers in the community would consider their generative model of molecules as a neural database for virtual screening, the idea of approximating the existing database is not very common.
- Quality: The model proposed is sound.

# Weaknesses
- Clarity: One weakness is the presentation issue. I had trouble reading the preliminaries section (Section 3.1), because i) many of the objects ($T$, $R$, $S$, $X$, $\mathcal{P}(\cdot)$, etc.) are not explicitly defined, and ii) technical concepts such as R-group, group, _multi-component_ reaction, and synthon chain are not defined in the paper (and I could not find "R-group" in reference [42]). Since the main audience of the NeurIPS conference is not fully aware of the terminology in drug discovery, it would be very helpful (at least to me!) to explain these concepts with some examples.
- Quality: The empirical studies are limited. Since CSLVAE will be used as a substitute for the original CSL, it is necessary to demonstrate its capability by applying CSLVAE to some realistic tasks to see the computational efficiency and potential performance degradation as compared to the baseline method using the original CSL. Without such an application, the neural database concept suggested in the paper cannot be appreciated.

---

> ### Author Response · Authors · 2022-08-02
> **Rebuttal to Reviewer UvPu**
>
> Thank you for your comments. We take to heart your request for clarity. We have rewritten and clarified our notation in this section to make the manuscript more self-contained. We are adding a figure to the supplementary materials to illustrate the concepts better. This will be reflected in our revision.
>
> Incidentally, in the cited reference [42], please search for "R group" (no hyphen). It appears 10 times in the text of the paper by our count (first appearance on page two, first paragraph of the section titled *The REAL Space virtual library*).
>
> We appreciate your comment regarding the limited empirical studies and the need to demonstrate a concrete application of CSLVAE towards a realistic task which can better illustrate the advantages of our approach. To this end, we refer you to the analoging exercise that we shared in our response to Reviewer 16PZ, as this reflects a concrete example of how CSLVAE can be used as a substitute for analoging a CSL with conventional methods. We report a three orders of magnitude improvement in analoging speed compared to Arthor, a highly optimized chemical similarity search tool that is commonly used by practitioners for analoging synthesis libraries and which can be considered a gold standard, and demonstrate that CSLVAE is capable of retrieving high quality analogs on its own.

---

> > ### Comment · Reviewer_UvPu · 2022-08-05
> > **Re: Rebuttal to Reviewer UvPu**
> >
> > I appreciate the authors to provide valuable responses to my concerns. I could find "R group" in the paper, and would be able to better understand the concept.
> >
> > ## Empirical demonstration
> > I appreciate the authors to design such a demonstration and provide the result. The proposed method achieves significant speed up with slight performance degradation as compared to Arthor, and this is a very good example of a neural database. Assuming that the result can be included in the final version of the manuscript, I would like to increase my score.

---

### Official Review · Reviewer_JV4f · 2022-07-06

**Rating:** 7
**Confidence:** 4
**Soundness:** 3 good
**Presentation:** 3 good
**Contribution:** 4 excellent

**Summary:**

Generative modelling of chemicals relies on generated molecules being valid and synthesizable. Often however, due to the size of combinatorial synthesis libraries, it is enough to look for hits in a library (also to forgo synthesizing a molecule oneself). Such libraries can often be (relatively) compactly described as the products of a set of starting reagents and reactions. The authors of this paper build a generative model of molecules in a large combinatorial synthesis library that embeds all molecules in a latent space.

The authors demonstrate that examples of graph generative models cannot fit the distribution of synthesizable sequences, with much of their support landing on molecules outside of the library. They show their method has a large support, generating many unique molecules, and supports a large part of the library, having a large likelihood on much of the molecules, all of this compared to examples of graph generative models.

The authors conclude by demonstrating that the latent space is continuous with respect to Tanimoto similarity, presumably making it good for semi-supervised learning.

**Questions:**

See weaknesses above.

**Limitations:**

The main limitation of the method is that it relies on a compact description of the library in terms of reactions and reagents. Expanding this method to the set of all commercially available reagents may be a challenge.
As well, should a group of reagents appear multiple steps in a reaction, the architecture of the model makes it likely that the same reagent will be selected for both steps. This hinders the ability of the model to fit some sets of molecules, such as polymers.
The authors acknowledge the above limitations in their work.

**Strengths And Weaknesses:**

Strengths:
1) The authors introduce a generative model that is guaranteed to output molecules in a given library.
2) The authors demonstrate that embeddings of molecules in the library are continuous and so the generative model can be used for optimization or semi-supervised molecular design.
3) The authors demonstrate that state of the art methods cannot be used to reliably generate within a library, motivating their model.
The method points to a novel approach to molecular design that is likely to be very useful to practitioners.

Weaknesses:
1) It is not checked that deep language models, which now are able to reach >90% synthesizability, can fit combinatorial libraries.
2) The primary weakness of the paper is the limited check that the latent space may be used for learning chemical features: comparing the performance of classifiers or optimization methods in the latent space of this models and the example graph-generative, and some language models would greatly strengthen the paper.

---

> ### Author Response · Authors · 2022-08-02
> **Rebuttal to Reviewer JV4f**
>
> We thank the reviewer for their comments and suggestions.
>
> Your question about whether deep language models are capable of fitting combinatorial synthesis libraries is well-taken. We note that the ability of language models to fit molecular databases was investigated in a prior study by [Arús-Pous et al. (2019)](https://jcheminf.biomedcentral.com/articles/10.1186/s13321-019-0341-z), who applied deep language models to GDB-13, [Blum and Reymond (2009)](https://pubs.acs.org/doi/10.1021/ja902302h). GDB-13 is a database of $10^9$ compounds formed by enumerating molecules up to 13 atoms of C, N, O, S, and Cl and satisfying simple chemical stability and synthetic feasibility rules. The authors trained on 0.1% of the total library, comparable to our approach, and find that their model covers only about 70% of the full library. More importantly, their calculation required approximately 750 hours of computation (as this involves both sampling a large number of compounds and then separately checking which among the sampled compounds are contained in the database). *All currently available deep language-based architectures will suffer from this limitation.* The CSL we consider is approximately 20 times larger, making such an analysis a significant computational challenge, and we do not believe such a comparison is technically feasible for even larger synthesis libraries presently available. We will cite the study and discuss challenges with applying current deep language models.
>
> As an additional note regarding the reviewer's comment that modern deep language models are able to achieve >90% synthesizability, we feel that it is worth emphasizing the important role CSLs play in early stage drug discovery, as they offer billions of drug candidates that are guaranteed to be synthesizable at low cost, low lead-time, and high throughput. For these reasons, CSLs are increasingly becoming the norm in early stage drug discovery pipelines. In other words, synthesizability alone is not sufficient for early stage drug discovery. To the best of our knowledge, our paper is the first generative model for enabling the navigation of such libraries and which can scale to their ever-increasing size. We believe that methods such as what we have proposed will prove to be important tools in enabling virtual screening to CSLs going forward.
>
> We appreciate the reviewer's comment regarding investigating the latent space's suitability for downstream tasks such as molecular property prediction. We note that the CSLVAE latent space is learned in a manner to support efficient querying from large combinatorial synthesis libraries, which is our primary objective. As mentioned in our response to Reviewer 16PZ, our validation confirms good performance for analog enumeration in comparison to an established baseline for analoging large synthesis libraries. Conventional tools like Arthor simply do not scale to the size of libraries that are increasingly becoming the norm in virtual screening; CSLVAE, on the other hand, is very well suited to such regimes given its logarithmic scaling due to the design of the decoder. In fact, we were required to limit our comparison to a library of $10^{10}$ compounds, rather than a larger $10^{12}$ library available to us to accommodate the limitations of Arthor. Exploration of molecular optimization based upon a generative latent space are extensive, but would still require significant additional effort and does not add to the primary story of a practical library analog enumeration method. Further, our purpose is in learning a latent space which permits access to large CSLs to support analoging tasks, rather than learning an encoder which can transfer to downstream predictive tasks, so we do not perform any such experiments as we do not believe it connects to our motivations with this work.

---

> > ### Comment · Reviewer_JV4f · 2022-08-03
> > **Response to author comment**
> >
> > Paragraphs one and two address my concerns about comparisons to language models well: under the author's model, one does not have to perform a costly calculation to see whether proposed molecules belong to a library of interest.
> >
> > Regarding my concern that the authors have not sufficiently demonstrated their ability to use the latent space for drug design, the authors point out that they demonstrate it use for analoging, which is significant enough. This is reasonable, as one can use any other chemo-informatics method to optimize a molecule and then analogize it into the library; this also has the benefit of the ability to learn from data outside of the library. However, other methods my propose molecules difficult to analogize. As well, the authors only are able to demonstrate that analogues are close in tanimoto similarity, not that they may not lose some property of interest. Thus I still believe it is a weakness of the paper that the authors have not sufficiently validated that applicability of their model to drug design.

---

> > > ### Author Response · Authors · 2022-08-08
> > > **Response to Reviewer JV4f**
> > >
> > > Thank you for the positive response.
> > >
> > > As a (final) remark to your comment that the analogues returned by CSLVAE, while perhaps close in Tanimoto similarity, may not preserve some properties of interest, we should note that this is true of other analogue enumeration approaches as well. The typical analouging workflow first retrieves compounds from the library that are sufficiently close, according to some measure of similarity, and then filters are applied post-hoc to the retrieved compounds to remove those which do not satisfy the specified criteria (e.g., logP, molecular weight). Hence, this exercise was intended as a concrete demonstration of how CSLVAE could be utilized in drug design, since analogue enumeration is a bottleneck with the growing sizes of combinatorial synthesis libraries, and there is a need for more efficient approaches for analouging in these regimes (to which Arthor is unable to scale). In fact, while this paper was under review, a new version of the REAL database was released that is again 50% larger, continuing a trend of rapidly growing CSLs.
> > >
> > > Nonetheless, we understand that the reviewer wishes for further demonstration that CSLVAE has utility in drug design settings. In particular, the reviewer points out that, "the primary weakness of the paper is the limited check that the latent space may be used for learning chemical features." We detail our efforts to address this concern via an additional experiment, described below.
> > >
> > > In this experiment, we sample 100K compounds from REAL and split these compounds into training, validation, and testing folds using an 80:10:10 split. For each compound, we extract their molecular queries from the CSLVAE encoder, in addition to calculating their ECFP4 and ECFP6 fingerprints. Using the training fold, we fit a multi-layer perceptron on each of these feature descriptors separately to predict the molecule's logP (cf. QED). We select the iteration which achieves the lowest validation RMSE and report the test set RMSE. For each input-target pair, we train five such models and report the average test RMSE and standard deviation. To demonstrate the extent to which CSLVAE learns molecular features that are predictive of such molecular properties for out-of-domain compounds, we take 250K molecules from ZINC and perform a similar experiment using an 80:10:10 split. The results of this exercise are summarized by the following table.
> > >
> > > |    | Number of features | REAL 100k logP | ZINC 250k logP | REAL 100k QED | ZINC 250k QED |
> > > | ----- | ----------- | ----------- | ----------- | ----------- | ----------- |
> > > | **CSLVAE** | 64 | **0.539 ± 0.002** | 0.591 ± 0.001 | **0.072 ± 0.001** | 0.068 ± 0.001 |
> > > | **ECFP4** | 256 | 0.827 ± 0.001 | 0.679 ± 0.002 | 0.091 ± 0.002 | 0.079 ± 0.001 |
> > > | **ECFP6** | 1024 | 0.601 ± 0.002 | **0.490 ± 0.001** | **0.072 ± 0.001** | **0.064 ± 0.001** |
> > >
> > > We note that the latent space learned by CSLVAE is indeed predictive of such molecular properties, both in-domain as well as out-of-domain. In all cases, it out-performs using the ECFP4 descriptors, and it performs favorably to ECFP6 descriptors within REAL (with ECFP6 leading to lower RMSEs in ZINC).
> > >
> > > Although our motivations are not in using CSLVAE as a pre-training technique for such downstream predictors, we feel that this experiment addresses the question raised by you as to whether the latent space can be used for learning chemical features in the affirmative.

---

### Official Review · Reviewer_16PZ · 2022-07-11

**Rating:** 5
**Confidence:** 2
**Soundness:** 2 fair
**Presentation:** 2 fair
**Contribution:** 2 fair

**Summary:**

This paper describes a method called CLSVAE that generates combinatorial libraries of small molecules using a continuous underlying representation. The authors demonstrate the forward and reverse query in such a library.


**Questions:**


Can the authors please make a compelling case for this model by demonstrating a usecase that would be onerous with a different toolchain and now becomes accessible?  In the absence of such a demonstration I don't know how to judge the significance of this work and thus feel compelled to reject it from the current conference.

How does the reverse search for synthons work when there are degenerate ways to synthesize the same molecule?  Could the authors provide a subset of statistical results for only such molecules?

[Edit: I've updated my score in part due to the additional effort by the authors.  The degeneracies can be significantly more common in large libraries than what the authors suggest.  Also, there are additional ways to find analogues rapidly, and even faster than the current speed the authors demonstrate.  However, the overall ideas are useful and might be worthy of publication.]

**Limitations:**

The authors address some of the possible technical limitations of this work.  There are no negative societal implications of this work.

**Strengths And Weaknesses:**


This work is original, is of reasonable quality, of mixed clarity, and of unclear significance.
The high level idea behind the paper appears reasonable, although the code is currently not provided so it is hard to judge the reproducibility or generalizability of this work in practice (the authors say that they would provide code if this work was accepted).  The description of the method is clear and the math makes sense to me.  The main weakness of this work, however, is that it does not appropriately motivate the rationale behind it so I cannot judge if this is actually significant in practice or not.

In the presence of the chemical reactions, the forward problem of generating a molecule from the given synthons is actually trivially accomplished with the current methods. The inverse problem of finding the synthons from the molecule is somewhat interesting, though it does not appear a priori difficult to solve with existing methods (though there might be more trial and error involved than with a differentiable code). Perhaps the main point of this work then is that there is some simplification in the search for similar molecules, e.g. during the phase of SAR by catalogue, though it is hard to imagine that such the exhaustive search would be difficult if one limited the library to only those with common disynthons to the full molecule and it is not clear how accurate the search in this latent space actually is. Although it is good to see comparisons with prior generic generators, obviously the main contribution of such molecular generators is that they can in principle access a space much richer than the limited space of enumerable combinatorial libraries (even if the latter soon will exceed 10^12 molecules in size).

---

> ### Author Response · Authors · 2022-08-02
> **Rebuttal to Reviewer 16PZ**
>
> Thank you for your request for a compelling use case that demonstrates its capabilities in settings that are not well-suited to existing tools. We designed an experiment to identify close analogs of 24 novel drugs approved in 2021 by the FDA--a common step in drug discovery. We compare CSLVAE with [Arthor from NextMove](https://www.nextmovesoftware.com/arthor.html), which is a state-of-the-art, commercial similarity search tool for synthesis libraries. We use each method to select 100 analogs from $10^{10}$ molecules in REAL and compute their ECFP4-based Tanimoto similarity to the query molecule, keeping the top-1 example. Because autoencoding with CSLVAE is a stochastic procedure, we repeat this 30 times, and average the top-1 Tanimoto similarities. As a baseline, we repeat this procedure using random sampling from REAL. In the table below, we report quantiles of the top-1 similarities computed over the 24 compounds. As a rule-of-thumb, compounds with Tanimoto similarities above 0.3-0.35 are typically considered analogs (we bold when the Tanimoto simliarity exceeds 0.35 for improved readability). CSLVAE performs above this threshold. While Arthor finds slightly closer analogs (median 0.48 vs. 0.37 for CSLVAE), **CSLVAE is more than three orders of magnitude faster**, owing to its logarithmic complexity in the size of the combinatorial synthesis library (which we formalize in section 3.4 of the paper). We believe that the quality of analogs found with CSLVAE can begin to approach those found by Arthor with further improvements. Our contribution with this paper is the development of a fast and efficient graph-based autoencoder for accessing such libraries, which can be further refined for specific applications like analoging. But we do not dedicate effort in this work to building a state-of-the-art analoging tool based on CSLVAE (as our focus is on the architecture itself and its computational advantages) and this exercise should be viewed as a demonstration that it is indeed possible to develop such applications using CSLVAE as a backbone.
>
> | | Min | 5% | 25% | 50% | 75% | 95% | Max | Time (sec.) |
> | ------ | ----- | ----- | ----- | ----- | ----- | ----- | ----- | ----- |
> | **Random** | 0.2590 | 0.2660 | 0.2961 | 0.3129 | 0.3289 | 0.3393 | **0.3691** | -- |
> | **Arthor** | 0.2674 | 0.3150 | **0.4309** | **0.4792** | **0.5525** | **0.6399** | **0.6774** | 30,216.70 |
> | **CSLVAE (ours)** | 0.2373 | 0.2551 | 0.3238 | **0.3687** | **0.4175** | **0.5457** | **0.7464** | 11.41|
>
> To your second question, we note the following. If there is a trivially degenerate synthesis route for a compound, i.e., if a reaction is exactly duplicated, then the representations for each reaction and all of their components are exactly the same. Therefore, all of the relevant choice probabilities are identical (a consequence of the inductive nature of the library encoder). Thus, the decoder assigns identical probabilities to such trivial degenerate synthesis routes. That said, there may certainly be instances in which a compound can be reached through distinct synthesis routes, and it may not be the case that the decoder assigns identical probabilities to the two routes. We will include an analysis of this in the supplementary materials. In any case, we do not believe this to be a significant practical limitation of our approach.

---

> ### Author Response · Authors · 2022-08-08
> **Please see Response to Reviewer JV4f for additional experiment**
>
> We wanted to make you aware of an additional experiment that we shared in our Response to Reviewer JV4f.

---

### Meta-Review · Area_Chair_uj5m · 2022-08-26

**Recommendation:** Accept
**Confidence:** Certain

**Metareview:**


There are three reviews, all of which place this paper above the threshold.  All reviewers agreed on several strong points, such as originality, convincing motivation behind the general approach and clarity of presentation. At the same time, the reviewers also raised some critical comments about the practical utility of the model (and a lack of experiments proving this utility). This last point of criticism, however, was addressed reasonably well in the rebuttal. Therefore, I think that the positive aspects dominate, and I recommend to accept this paper.

**Award:**

No

---

### Decision · Program_Chairs · 2022-09-14

Accept